# DoxoDB: A Database for the Expression Analysis of Doxorubicin-Induced lncRNA Genes

**DOI:** 10.3390/ncrna9040039

**Published:** 2023-07-13

**Authors:** Rebecca Distefano, Mirolyuba Ilieva, Jens Hedelund Madsen, Sarah Rennie, Shizuka Uchida

**Affiliations:** 1Department of Biology, University of Copenhagen, DK-2200 Copenhagen N, Denmark; fwh492@alumni.ku.dk; 2Center for RNA Medicine, Department of Clinical Medicine, Aalborg University, DK-2450 Copenhagen SV, Denmark; mirolyubasi@dcm.aau.dk (M.I.); jenshm@dcm.aau.dk (J.H.M.)

**Keywords:** cancer, cardiovascular disease, doxorubicin, RNA-seq

## Abstract

Cancer and cardiovascular disease are the leading causes of death worldwide. Recent evidence suggests that these two life-threatening diseases share several features in disease progression, such as angiogenesis, fibrosis, and immune responses. This has led to the emergence of a new field called cardio-oncology. Doxorubicin is a chemotherapy drug widely used to treat cancer, such as bladder and breast cancer. However, this drug causes serious side effects, including acute ventricular dysfunction, cardiomyopathy, and heart failure. Based on this evidence, we hypothesize that comparing the expression profiles of cells and tissues treated with doxorubicin may yield new insights into the adverse effects of the drug on cellular activities. To test this hypothesis, we analyzed published RNA sequencing (RNA-seq) data from doxorubicin-treated cells to identify commonly differentially expressed genes, including long non-coding RNAs (lncRNAs) as they are known to be dysregulated in diseased tissues and cells. From our systematic analysis, we identified several doxorubicin-induced genes. To confirm these findings, we treated human cardiac fibroblasts with doxorubicin to record expression changes in the selected doxorubicin-induced genes and performed a loss-of-function experiment of the lncRNA *MAP3K4-AS1*. To further disseminate the analyzed data, we built the web database DoxoDB.

## 1. Introduction

Cardiovascular disease and cancer rank as the first and second leading causes of death worldwide, respectively [1]. Although there are numerous etiological differences between the two conditions, significant connections have emerged, particularly in cancer survivors [2,3]. While cancer fatality rates have significantly decreased over the past three decades, the cardiotoxicity resulting from certain cancer treatments is now recognized as the leading cause of mortality in oncology patients [4,5]. The identification of this relationship has led to the development of a new field called “cardio-oncology”, which focuses on preventing and treating adverse cardiac events arising from cancer therapies [2,4,6]. The emergence of cardio-oncology has also uncovered several shared disease progression mechanisms between cancer and cardiovascular disease, such as angiogenesis, fibrosis, and immune responses [7].

Doxorubicin (DOX) is an anticancer agent belonging to the anthracycline family. It is commonly employed in several pediatric cancer treatments and against various tumor types, including leukemia, bladder, breast, gastrointestinal, lung, and stomach cancer. Its primary anticancer mechanism involves DNA interaction and inhibition of the topoisomerase II enzyme, ultimately resulting in cancer cell death [3,4,8]. While DOX has been used since the 1970s due to its broad spectrum of anticancer activity, the development of chronic and acute cardiovascular complications has been observed in approximately 50% of pediatric cancer survivors and in adult oncology patients [8,9]. Its cumulative and dose-dependent cardiotoxicity ranges from subtle changes in the function and structure of the heart to severe cardiomyopathy and congestive heart failure, which may ultimately lead to death or cardiac transplantation [2,10,11]. Notably, DOX toxicity has been shown to affect various cardiac-associated cell types, including cardiomyocytes, cardiac fibroblasts (CFs), and endothelial cells [2,12]. Although previous research has primarily focused on cardiomyocytes [11,13,14,15,16,17], the impact of DOX on CFs has received limited attention. Given that CFs play a crucial role in cardiac growth, injury response, and myocardial stress response [18,19,20], they would be expected to serve as a natural focus in the understanding of DOX toxicity. However, the underlying mechanisms of DOX-induced cardiotoxicity remains incomplete, and no effective treatment has been found to date [2].

Non-coding RNAs (ncRNAs) have become a subject of increased interest in recent years due to their role in the pathogenesis of various diseases [21]. ncRNAs account for a large portion of the mammalian genome and include ribosomal RNAs (rRNAs), transfer RNAs (tRNAs), small RNAs (e.g., microRNAs (miRNAs)), and long ncRNAs (lncRNAs), where the latter are often defined in the literature as ncRNAs longer than 200 nucleotides [22,23]. LncRNAs are abundant in mammalian genomes, and their cell-specific expression patterns have prompted extensive investigation into their functions, which include imprinting, epigenetic, transcriptional, post-transcriptional, and translational regulation [24]. Studies have documented the expression of various lncRNAs in the development and maintenance of the cardiovascular system, as well as their roles in numerous diseases, including cardiovascular disease and cancer [14,25,26,27]. Furthermore, some lncRNAs have been implicated in cancer cell drug resistance and the regulation of cancer cells’ response to chemotherapy drugs, including DOX [28,29]. For instance, lncRNAs, such as differentiation antagonizing non-protein-coding RNA (*DANCR*) [30], damage-induced long non-coding RNA (*DINOL*) [31], prostate-specific transcript (*PCGEM1*) [32], and p53 up-regulated regulator of p53 levels (*PURPL*) [33] have been linked to the regulation of DOX-induced apoptosis in cancer cells. Another example is cardiomyocyte mitochondrial dynamic-related lncRNA 1 (*CMDL-1*), which has been linked to an antiapoptotic role in DOX-induced cardiotoxicity in cardiomyocytes [13]. In addition, dysregulation of long intergenic non-protein-coding RNA 39 (*LINC00339*) [14], *NONMMUT015745* [34], and SOX2 overlapping transcript (*SOX2-OT*) [35] has been associated with DOX-induced apoptosis in cardiomyocytes, while dysregulation of metastasis-associated lung adenocarcinoma transcript 1 (*MALAT-1*) [36] and tumor protein p53 pathway corepressor1 (*Trp53cor1*, also known as *lincRNA-p21*) [37] has been related to cardiac cellular senescence.

Although several lncRNAs have been identified and characterized in cancer cells and cardiomyocytes, less is known about DOX toxicity in fibroblasts, where only two lncRNAs have been studied. These include *lincRNA-p21*, which has been reported to mediate p53-mediated cell cycle arrest in DOX-treated mouse embryo fibroblasts [38], and *PCGEM1*, whose elevated expression levels have been linked to an increase in colony formation on LNCaP prostate cancer cells and in NIH3T3 mouse fibroblasts cells [32]. This observation is of particular interest as a recent study demonstrated that DOX induces mitochondrial dysfunction in CFs [9]. Given that CFs play an integral role in myocardial growth and homeostasis, as well as in response to stressors and injury [39], further research is required to understand the cardiotoxicity induced by DOX in CFs.

The primary objective of this study is to identify DOX-induced genes, with a particular emphasis on lncRNA genes. To this end, we systematically re-analyzed four published RNA-seq datasets from DOX-treated cells to identify commonly shared DOX-induced lncRNA genes across a broad range of relevant cell types. Given the lack of knowledge of DOX’s effects on CFs and DOX-induced lncRNAs, we validated the expression profiles of four identified DOX-induced lncRNA genes in human CFs treated with DOX and carried out loss-of-function experiments to understand the potential functional role of one of these lncRNA genes: *MAP3K4* antisense RNA 1 (*MAP3K4-AS1*). Finally, we created a web database, DoxoDB, intending to disseminate the expression profiles of DOX-induced protein-coding and lncRNA genes to the scientific community (https://rebeccadistefano.shinyapps.io/DoxoDB/ (accessed on 3 July 2023)).

## 2. Results

### 2.1. Differentially Expressed Genes in Human Pulmonary Fibroblasts Treated with DOX

Pulmonary arterial hypertension (PAH) is caused by high blood pressure in the lungs. The prevalence of PAH is increasing, with approximately 10.6 cases per 1 million adults in the United States [40]. PAH can be caused by pulmonary fibrosis, where scarred lung tissues result in right ventricular hypertrophy and right-sided heart failure [41], and is thus highly relevant to cardiovascular disease. To investigate alterations in the expression of genes related to DOX treatment, previously published RNA-seq data of pulmonary fibroblasts treated with DOX were re-analyzed (Gene Expression Omnibus (GEO), accession number GSE154101). The original study aimed to investigate a vasculo-regenerative program in pulmonary endothelium mediated by the PPARγ-p53 transcription factor complex and demonstrated the potential role of a novel p53-based therapeutic strategy for PAH [42]. The RNA-seq data were generated from pulmonary artery adventitial fibroblasts (PAAFs) treated with DOX to increase p53 activity, and the control cells treated with phosphate-buffered saline (PBS). Since lncRNAs were not explored in the original study, we analyzed them in conjunction with the protein-coding genes below.

First, we examined the expression levels of protein-coding and lncRNA genes, which showed that the expression of lncRNA genes was generally lower compared to that of protein-coding genes (Figure 1A), as expected on the basis of previous studies [43,44]. Next, we conducted a differential expression analysis between DOX-treated PAAF cells and control cells. Based on the threshold values of 2-fold (fold change in logarithm of base 2 (|log_2_FC|) > 1) and the false discovery rate-adjusted *p*-value (FDR) of <0.05, 1565 up- and 161 down-regulated genes were identified (Appendix A). Of these, 492 and 3 were up- and down-regulated lncRNA genes, respectively (Figure 1B,C), with lncRNA genes showing a tendency to be specifically up-regulated relative to protein-coding genes (*p*-value < 0.00001, Fisher’s exact test), indicating that there may be some lncRNA genes related to the pathways/processes preferentially being up-regulated.

To categorize differentially expressed genes (DEGs), we analyzed protein-coding genes for enriched Gene Ontology (GO) terms and Kyoto Encyclopedia of Genes and Genomes (KEGG) pathways. Several GO terms were found to be enriched in the list of up-regulated protein-coding genes (Figure 1D), which were related to changes in tissue remodeling and cell signaling, as well as in chromatin remodeling and nucleosomes. These DEGs are likely linked to DOX’s ability to act as a DNA intercalator to cause DNA double-stranded breaks, which has been shown to increase nucleosome turnover [45]. Furthermore, along with the KEGG pathways involved in immune responses and p-53 signaling pathways, the pathways “*systemic lupus erythematosus*”, “*alcoholism*”, and “*neutrophil extracellular trap formation*” (Figure 1E) were also found to be enriched. Interestingly, these three enriched pathways contained histone genes (H2A, H2B, H2C, and H4C histone families), reflecting the release of histones and the eviction of nucleosomes caused by DOX’s influences on DNA intercalation [46,47]. While no KEGG pathway was significantly enriched in the list of down-regulated protein-coding genes, GO analysis resulted in five significantly enriched GO terms involved in intracellular structures (Figure 1F). Thus, the analysis indicates that DOX-induced protein-coding genes are associated with changes in tissue remodeling, cell signaling, chromatin remodeling, and intracellular structure.

Finally, to infer the potential functional roles of the differentially expressed lncRNA genes, we performed a GO analysis of the protein-coding genes that were most significantly correlated in their expression levels to the up-regulated lncRNA genes. This analysis revealed the enrichment of GO terms related to cell adhesion and tissue remodeling (Figure 1G) and KEGG pathways related to immune responses (Figure 1H). These results suggest that the DOX-induced lncRNA genes may function in the regulation of intra and extracellular structures as well as in chromatin remodeling and immune responses.

### 2.2. Differentially Expressed Genes in Human Foreskin Fibroblasts Treated with DOX

To further understand the genes induced by DOX, we performed a secondary analysis of the previously published RNA-seq data from the DOX-treated primary human foreskin fibroblasts (HFFs) [48] (GEO, accession number GSE135842). The aim of the original study was to investigate the differential effects of the retinoblastoma protein 1 (*RB1*), the RB transcriptional corepressor like 1 (*p130*), and the RB transcriptional corepressor like 1 (*p107*) on the expression of genes at different stages of the cell cycle following the activation of tumor protein p53 (*TP53*) by DOX-induced cell damage [48]. The RNA-seq data were generated from the HFFs with CRISPR/Cas9-mediated knockout of the *p130* gene alone (designated as sgP130 hereafter) or in combination with the *RB1* gene (designated as sgP130+sgRB1 hereafter). Furthermore, these cells were transfected with either siRNA against the *p107* gene (designated as siP107 hereafter) or against random sequences (designated as siCTRL hereafter). As a result, the RNA-seq dataset consists of the following four conditions, each with four replicates: sgP130_siCTRL, sgP130_siP107, sgP130+sgRB1_siCTRL, and sgP130+sgRB1_siP107. Two replicates of each cell type were treated with DOX (designated as 350 hereafter), while the remaining two were left as controls (designated as 0 hereafter), resulting in a total of 16 samples. The study demonstrated the specific roles of *RB1*, *p130*, and *p107* in a p53- and p21-mediated repression of the cell cycle, while also identifying novel gene targets for each of the genes [48].

To facilitate the detection of dysregulated lncRNA genes in this dataset, we conducted a differential expression analysis similarly to the previous subsection (2-fold and FDR < 0.05), identifying hundreds of DEGs in DOX-treated cells compared to control cells across the four different conditions (Figure 2; Appendix A). Given the large number of up- and down-regulated genes, we asked how many DEGs were shared among the four comparisons. While 115 and 23 genes were found up- and down-regulated in all four conditions, respectively, most of the DEGs were unique to each comparison (Appendix A). Among the shared genes, 16 out of the 115 up-regulated DEGs were lncRNA genes, indicating that lncRNA genes are dysregulated in DOX-treated HFFs compared to the control cells, regardless of the expression of the p107 and RB1 genes. To understand the shared genes better, protein-coding genes were further analysed for the enrichment of GO terms and KEGG pathways. While no KEGG pathway nor GO term was found to be significantly enriched in the shared up-regulated protein-coding genes, the GO analysis for the shared down-regulated protein-coding genes resulted in the enrichment of several GO terms related to the cell cycle and division (e.g., cell division (GO:0051301), midbody (GO:0030496), and spindle pole (GO:0000922)) (Appendix A). Consistently, the KEGG pathway “Oocyte meiosis” was also found to be significantly enriched in the set of shared down-regulated protein-coding genes (FDR of 0.0182). Thus, it is plausible that DOX-induced lncRNA genes may also be involved in the regulation of cell cycle progression.

### 2.3. Differentially Expressed Genes in Human Senescent Cells Treated with DOX

The accumulation of senescent cells during aging has been identified as a major contributor to excessive inflammation and imbalance in tissue homeostasis, and has been linked to various age-related diseases, including CVD and cancer [49]. Moreover, while cellular senescence is a well-recognized response to chemotherapy drugs, it has attracted attention for its connection to chemotherapy-induced cardiotoxicity. Consistently, the removal of senescent cells has been correlated with a reduction in the pathophysiology of cardiovascular disease and with the prevention of myocardial dysfunction in relation to chemotherapy-induced cardiotoxicity [15,50,51,52,53,54]. Furthermore, previous studies have reported DOX’s ability to induce cellular senescence in fibroblast cells, including CFs [55,56]. Given this link between anthracycline-induced senescence and chemotherapy-induced cardiotoxicity, we next aimed to identify DOX-induced genes in human senescent cells to better understand DOX-induced cardiotoxicity by analyzing the previously published RNA-seq data of DOX-induced senescent HFF HCA2 cells (GEO, accession number GSE198396). The original study aimed to elucidate the role of the immune system by focusing on the accumulation of senescent cells during natural aging through the programmed cell death ligand 1/programmed cell death 1 (PD-L1/PD-1) immune checkpoint pathway [49]. The RNA-seq dataset was derived from proliferating HCA2 cells (used as a control), PD-L1-positive (referred to as PD-L1+ hereafter), and PD-L1-negative (referred to as PD-L1- hereafter) DOX-induced senescent HCA2 cells, with each group in triplicate. The study reported that blocking the PD-L1/PD-1 pathway resulted in a reduction in age-related inflammation and enhanced clearance of senescence cells [49].

Once again with the aim of analyzing dysregulated lncRNA genes, we conducted a differential expression analysis identifying 2113 up- and 1169 down-regulated genes in PD-L1+ DOX-induced senescent cells compared to the control cells. Among these DEGs, 505 up- and 101 down-regulated were lncRNA genes (Figure 3A,B; Appendix A). Similarly, 1731 up- and 1286 down-regulated genes were identified in PD-L1- DOX-induced senescent cells compared to the control cells, with 325 and 141 of these being up- and down-regulated lncRNA genes, respectively (Figure 3A,B; Appendix A). It is evident that the expression of lncRNA genes is altered in DOX-induced senescent cells, hinting that lncRNA genes may participate in cellular senescence. Notably, no DEG was identified in PD-L1+ cells compared to PD-L1- cells.

Considering the large number of DEGs in senescent cells compared to controls, we next examined the overlap between the two comparisons. We identified several common up- and down-regulated genes between PD-L1+ and PD-L1- cells in comparison to the control cells (Figure 3C; Appendix A). The GO analysis of shared up-regulated protein-coding genes revealed the enrichment of GO terms related to the modification of the microenvironment involved in senescence and to the senescence-associated secretory phenotype (SASP) (Figure 3D), characterized by the release of chemokines, inflammatory cytokines, proteases (including matrix metalloproteinases), and growth factors, which collectively contribute to the promotion of chronic inflammation [57,58,59,60,61,62]. Additionally, the KEGG pathways analysis revealed the enrichment of pathways involved in immune responses and the pathways “systemic lupus erythematosus”, “alcoholism”, and “neutrophil extracellular trap formation” (Figure 3E), which consist of histone genes (H2A, H2B, H2C, and H4C histone families). These reflect the changes in chromatin dynamics as a result of DOX’s influence on DNA intercalation [46,47], in line with the results of the previous dataset of PAAF cells (Section 2.1). Moreover, the enriched GO terms for shared down-regulated protein-coding genes primarily indicated a reduction in cell division (Figure 3F), which is in line with the results obtained in the four HFF conditions described above (Section 2.2) (Appendix A). DOX-induced cell cycle arrest [63] was also evident in the significantly enriched KEGG pathways, which were mostly involved in cell cycle progression (Figure 3G).

Inspired by the above analysis, we further dissected the differentially expressed lncRNA genes by examining the expression correlation between protein-coding and lncRNA genes. While the analysis of down-regulated lncRNA genes resulted in the enrichment of GO terms and KEGG pathways involved in the cell cycle (Appendix A), the analysis of up-regulated lncRNA genes revealed the significant enrichment of GO terms involved in SASP (Appendix A), and eight KEGG pathways associated with protein degradation (Appendix A). These included the pathways “proteosome” as well as “amyotrophic lateral sclerosis”, “spinocerebellar ataxia”, “pathways of neurodegeneration-multiple disease”, “Parkinson’s disease”, and “Alzheimer’s disease”, which primarily involved genes associated with the proteasome machinery. Notably, proteosome activity has been observed to decrease in senescent cells [64], while DOX treatment has been shown to increase proteasome activity in cardiomyocytes, leading to the development of pressure overload cardiac hypertrophy and remodeling [65,66].

These results are consistent with the above analyses and further support the idea that the top-correlating protein-coding genes may have a similar function to the DOX-induced lncRNA genes. Thus, it is likely that DOX-induced differentially expressed lncRNA genes could also be involved in SASP, tissue remodeling, histone modulations, and in cell cycle progression, consistent with the results highlighted in the above subsections.

### 2.4. Differentially Expressed Genes in DOX-Resistant Inflammatory Breast Cancer Cells

Based on the results highlighted above, it is possible that the identified DOX-induced DEGs are involved in SASP and immune responses. In particular, SASP is characterized by the secretion of proinflammatory cytokines, and studies have shown that therapy-induced senescence could lead to chronic inflammation and drug resistance [52]. In addition, systemic inflammation has previously been demonstrated as one of the potential mechanisms that contributes to the development of DOX-induced cardiotoxicity [67,68,69,70]. Hence, to investigate the relationship between DOX-induced cardiotoxicity and inflammation, we performed secondary analysis of the previously published RNA-seq data of DOX-treated inflammatory breast cancer (IBC) cells (GEO, accession number GSE163361). The original study aimed to investigate chemotherapy resistance mechanisms in IBC, a disease known for its poor clinical outcome and elevated risk of metastasis and treatment resistance [71]. The RNA-seq dataset was generated from two IBC cell lines: SUM149 and FCIBC02. Both parental and DOX-resistant cells were included in the analysis. Additionally, these cells were treated with either vehicle or DOX, with each group in duplicate. This resulted in a total of eight samples per cell line: parental control cells, parental DOX-treated cells, DOX-resistant control cells, and DOX-resistant DOX-treated cells. The study identified the Janus kinase 2/signal transducer and activator of transcription 3 (JAK2/STAT3) pathway as the main driver of chemotherapy resistance in IBC, along with the possibility of reversing this effect through a combination of paclitaxel and JAK2 inhibitors [71].

By applying the same threshold values as the previous analysis, 1407 up- and 1138 down-regulated genes (including lncRNA genes) were identified in FCIBC02 parental DOX-treated cells compared to control cells (Figure 4A,B; Appendix A). Moreover, 3641 up- and 1557 down-regulated genes were identified in SUM149 parental DOX-treated cells compared to the control cells (Figure 4A,B; Appendix A). Of these DEGs, 471 up- and 173 down-regulated genes were found to be shared among these two cell lines, while the remaining DEGs were unique to each IBC cell type. Among the shared genes, 87 and 27 were up- and down-regulated lncRNA genes, respectively (Figure 4C; Appendix A). Interestingly, the previously mentioned lncRNA, SOX2-OT, was found among the shared up-regulated lncRNA genes. This lncRNA gene has been demonstrated to be dysregulated in a wide variety of cancers, including breast cancer [72,73]. While the list of shared down-regulated protein-coding genes did not yield any significant result in the GO and KEGG pathway analysis, the list of shared up-regulated protein-coding genes was associated with GO terms related to changes in cell adhesion, cell signaling, and tissue remodeling (Figure 4D), similar to the results obtained in the previous subsections. Furthermore, the GO terms related to immune responses (e.g., “response to virus” (GO:0009615), “innate immune response” (GO:0045087), and “inflammatory response” (GO:0006954)) were also identified as being significantly enriched, indicating an up-regulation of immune response pathways as suggested by the original study [71] (Figure 4D). However, no significantly enriched KEGG pathway was identified in the list of shared up-regulated protein-coding genes.

The set of shared up-regulated lncRNA genes was further explored via a GO and KEGG analysis of the protein-coding genes most correlated in their expression. In line with the results obtained for the set of dysregulated protein-coding genes, this analysis revealed the enrichment of GO terms and KEGG pathways involved in tissue remodeling, the extracellular matrix, and immune responses (Figure 4E,F), consistent with the results obtained in the previous sections.

Next, by performing a differential expression analysis between parental control cells and DOX-resistant control cells, we identified DEGs not induced by DOX for both cell lines, finding 874 up- and 777 down-regulated genes in FCIBC02 cells (Appendix A), while 1352 up- and 971 down-regulated genes were identified in SUM149 cells lines (Appendix A). These DEGs included both protein-coding and lncRNA genes. Of these DEGs, 118 up- and 36 down-regulated genes were shared between these two cell lines, including 24 up- and 4 down-regulated lncRNA genes, respectively (Appendix A).

Taken together, our analyses further suggest that the identified DOX-induced differentially expressed lncRNA genes might be involved in tissue remodeling, as well as in the regulation of immune responses. Furthermore, the identification of DEGs not induced by DOX could serve as a valuable resource for refining the list of candidate DOX-induced DEGs. By comparing the list of DOX-induced DEGs in DOX-sensitive cells to that of DOX-induced DEGs in DOX-resistant cells, genes that are not specifically regulated by DOX can be filtered out, thus increasing the specificity of the analyzed results.

### 2.5. Commonly Dysregulated lncRNA and Protein-Coding Genes

The above analyses identified many dysregulated lncRNA and protein-coding genes in response to DOX treatment in the various cell types studied. The GO and KEGG analyses consistently showed an involvement of these DEGs in histone modulations, cell cycle progression, SASP, and immune responses. These results suggest that the identified lncRNA genes may also participate in these cellular activities as discussed above. To further explore this possibility, we conducted a detailed examination of the DEGs from each cell type to identify the shared differentially expressed lncRNA and protein-coding genes (Figure 5). While no lncRNA gene was found to be shared among all nine gene sets, the following four lncRNA genes were found consistently up-regulated in eight, six, and five out of the nine gene sets (Figure 5A): C1QTNF1 antisense RNA 1 (*C1QTNF1-AS1*), the novel transcripts (*ENSG00000272468* and *ENSG00000260912*), and *MAP3K4-AS1* (Appendix A).

*C1QTNF1-AS1* is encoded on chromosome 17: 79,016,664–79,027,673 with three isoforms [74]. In hepatocellular carcinoma, the overexpression of C1QTNF1-AS1 was shown to regulate *miR-221-3p* and suppressor of cytokine signaling 3 (*SOCS3*), leading to a decrease in cancer cells’ proliferation, migration, and invasion, while also inducing apoptosis via the JAK/STAT signaling pathway [75,76]. Furthermore, *C1QTNF1-AS1* has been linked to poor prognosis in cervical cancer patients [77] and has also been found to be one of the most abundantly altered genes in triple negative breast cancer cells in response to DOX treatment [78]. The novel transcript *ENSG00000260912* (also known as *RP11-363E7.4*) is located on chromosome 9: 19,453,207–19,455,171 with one isoform overlapping the protein-coding gene alkaline ceramidase 2 (*ACER2*) gene [74]. It has been reported to be involved in fibrogenesis in idiopathic pulmonary fibrosis [79] and found to be dysregulated in patients with myocardial infarction and atrial fibrillation [80,81]. Furthermore, studies have reported its dysregulation in two colon carcinoma cell lines (SW480 and Caco2) upon knockdown of the aurora kinase A (*AURKA*) gene [82], in papillary thyroid carcinoma [83], in hepatocellular carcinoma treated with cisplatin [84], and in gastric cancer [85,86]. Another novel transcript *ENSG00000272468* (also known as *Lnc-HIST1H2BJ-3*) [87] and *MAP3K4-AS1* [88] were reported to be dysregulated in stomach adenocarcinoma and in breast cancer, respectively. *ENSG00000272468* is located on chromosome 6: 27,122,657–27,123,221 with only one isoform, while *MAP3K4-AS1* is encoded on chromosome 6: 160,990,317–160,992,418, with one transcript. *MAP3K4-AS1* overlaps the protein-coding gene mitogen-activated protein kinase 4 (*MAP3K4*) gene [74]. Although these lncRNA genes have been studied in relation to cancer and cardiovascular disease to some extent, to the best of our knowledge, there is currently no research exploring the role of these three lncRNA genes specifically in the context of DOX-induced cardiotoxicity.

In addition, some protein-coding genes were found up-regulated in eight out of the nine gene sets (Figure 5B; Appendix A). These include claudin 1 (*CLDN1*), H2B clustered histone 8 (*H2BC8*), and Ras related glycolysis inhibitor and calcium channel regulator (*RRAD*). *CLDN1* is located on chromosome 3: 190,023,490–190,040,264, and has two transcripts, and it is normally involved in maintaining cell polarity in epithelial and endothelial cells [89]. Several studies have reported an increased expression of *CLDN1* in human lung adenocarcinoma, linking it to anthracycline (including DOX) resistance and decreased penetration [90,91], and in several other tumors, including colon, hepatocellular, and pancreatic adenocarcinoma, as well as in breast cancer [92,93,94,95]. Similarly, aberrant expression of *RRAD* has been linked to different tumors, with *RRAD* acting as a tumor suppressor or oncogene, depending on the cancer type [96]. *RRAD* is located on chromosome 16: 66,955,582–66,959,547 and has five transcripts [97]. By encoding for a protein that acts as a calcium channel regulator in cardiac L-type Ca^2+^ channels, RRAD plays an important role in the normal heart physiology and has been found highly expressed in cardiomyocytes [98,99,100] with RRAD deficiency observed in failing human hearts [101]. Murine studies have also highlighted the link between decreased *RRAD* expression and cardiac hypertrophy in vitro [101], which was subsequently confirmed in human embryonic stem cell line (hESCs)-derived cardiomyocytes [98], along with cardiac fibrosis in vivo [102]. Finally, *H2BC8* is located on chromosome 6: 26,216,200–26,216,688 and has only one transcript. This gene encodes a protein that is involved in chromatin dynamics. Although no specific studies have been conducted on *H2BC8* regarding its association with DOX-induced cardiotoxicity, a study demonstrated the significant impact of DOX on the intracellular distribution of H2B-H2A dimer components. This study revealed the translocation of H2B from the nucleus and its notable accumulation in the cytoplasm in Jurkat and human peripheral blood mononuclear cells, resulting in an unstable nucleosome [103]. Furthermore, another study reported the dysregulation of *H2B* genes in breast cancer cell lines resistant to epirubicin and DOX [104]. Taken together, similar to the identified lncRNA genes, these protein-coding genes have not been investigated in the context of DOX-induced cardiotoxicity.

### 2.6. Loss-of-Function Experiments in Human Cardiac Fibroblasts

Compared to other cell types in the heart (e.g., cardiomyocytes), the impact of DOX on CFs is poorly understood. To address this gap in knowledge, we treated CFs with varying amounts of DOX for 24 h (1, 10, and 100 nM) (Figure 6A). As expected, the higher concentrations of DOX resulted in cell death as DOX accumulated in mitochondria [105]. Due to the high toxicity of 10 and 100 nM DOX resulting in dying cells, we treated CFs with 1 nM of DOX and measured the expression of DOX-induced protein-coding genes (*CLDN1, RRAD,* and *H2BC8*) and lncRNA genes (*C1QTNF1-AS1, ENSG00000260912, ENSG00000272468,* and *MAP3K4-AS1*) in comparison to the non-treated CFs as control (Figure 6B). As expected, we observed an up-regulation in the expression of all protein-coding and lncRNA genes, which further confirmed the selection made in the previous subsection.

To test whether these DOX-induced lncRNA genes have any functional role in CFs, we silenced the expression of *MAP3K4-AS1* using siRNAs. Compared to the control siRNA (siRNA against scrambled sequences), the expression of *MAP3K4-AS1* was significantly reduced upon silencing with three siRNAs against this lncRNA gene (Figure 6C). According to the latest Ensembl annotations (GRCh38.109) [74], there is only one transcript (thus, no other isoforms) for this lncRNA gene. As shown in Figure 6A, DOX induces apoptosis in CFs. Thus, we treated siRNA-transfected CFs with 1 nM of DOX and recorded the cellular viability, cytotoxicity, and apoptosis upon silencing of *MAP3K4-AS1* (Figure 6D–F). Compared to the control, *MAP3K4-AS1*-silenced CFs showed increased cytotoxicity and apoptosis, suggesting that *MAP3K4-AS1* might play a protective role against DOX in CFs. However, further functional, and mechanistic studies are necessary to understand the exact role of *MAP3K4-AS1* in CFs and under DOX treatment.

### 2.7. The Web Database for Protein-Coding and lncRNA Genes Induced by DOX: DoxoDB

To disseminate the results from the current study and to facilitate further research on DOX-induced genes, we built the database DoxoDB (Figure 7A). This is a user-friendly web application that can be either accessed online at https://rebeccadistefano.shinyapps.io/DoxoDB/ (accessed on 3 July 2023) or run locally using R (https://github.com/Reb08/DoxoDB/ accessed on 4 June 2023). DoxoDB contains the expression profiles of all the protein-coding and lncRNA genes analyzed above, providing a convenient platform to perform the expression analysis of condition-specific (e.g., DOX-induced cell cycle regulation or senescence) expression changes.

DoxoDB has four main pages: Explore, Download, lncRNAs, and Documentation. In the Explore page, the users can easily explore expression changes of protein-coding and lncRNA genes in the different DOX-treated cells analyzed above. The results are conveniently displayed in the “Result Table” located on the left-hand side of the page (Figure 7B). This page is dynamically generated according to the user-defined fold change (log_2_FC) and FDR values, which control the number of DEGs identified in each study. The DEGs can be further examined through a volcano plot and heat map (Figure 7B). The volcano plot is directly linked to the “Result Table”, enabling users to select a specific gene from the table, which will be highlighted on the plot. Furthermore, DEGs can be analyzed for enriched GO terms and KEGG pathways, differentiating between up- and down-regulated genes (Figure 7C). Finally, given the different comparisons available for each study, the number of shared DEGs across the different comparisons can be visualized on the “Comparisons Intersection” window within the Explore page through a Venn diagram. A text output listing the shared DEGs is also provided (Figure 7D).

The lncRNA genes identified in this study can be further explored in the lncRNAs page, where the lncRNA Table displays the lncRNA genes selected by applying the threshold values of 2-fold and FDR < 0.05 from each study and comparison. Here, the information about the conservation status [106], miRNA targets [106], terms associated with genome-wide association studies (GWAS) [106], nearest protein-coding gene, and the top correlated gene in its expression to the selected lncRNA gene can be found (Figure 7E). To facilitate further analysis by the users, the four datasets used in this study can be downloaded from the Download page. Lastly, the detailed instructions on how to use the database, information on the utilized datasets, and guidelines on providing feedback can be found on the Documentation page.

## 3. Discussion

As both cardiovascular disease and cancer are the leading causes of death worldwide, we aimed to uncover the expression patterns of lncRNA genes in these devastating diseases. The reason for focusing on DOX is that it is a widely used chemotherapy drug for various types of cancer and is well known as causing cardiac events as side effects. To this end, we identified genes induced by DOX through secondary analyses of four published RNA-seq datasets with a particular emphasis on lncRNA genes that were not considered in the original studies. Upon examining each dataset, we observed a common theme whereby DOX treatment led to the up-regulation of protein-coding genes linked with tissue remodeling, immune responses, and chromatin regulation, as well as a consistent down-regulation of genes implicated in cell cycle progression. These findings align with the goals and results of the original studies. By performing correlation analyses between protein-coding genes and differentially expressed lncRNA genes, we found that DOX-induced lncRNA genes may also participate in the same regulatory processes as the protein-coding genes. However, most of the identified dysregulated lncRNA genes have not yet been functionally or mechanistically studied. Moreover, none of them have been investigated in the context of DOX effects in CFs.

Among the lncRNA genes up-regulated across most of the conditions analyzed in this study, we focused on *MAP3K4-AS1*. This lncRNA has only been previously reported to be linked with a specific molecular subtype of breast cancer [87]. However, its role in the context of DOX-induced cardiotoxicity remains unexplored. To address this gap, we performed an expression analysis of DOX-treated CFs, where *MAP3K4-AS1* was found to be significantly up-regulated. Through knockdown experiments, we further showed that *MAP3K4-AS1* might play a protective role against DOX-induced cardiotoxicity by mitigating cytotoxicity and apoptosis in CFs. Nonetheless, further experiments are required to elucidate the functional and mechanistic roles of *MAP3K4-AS1* in DOX-induced cardiotoxicity.

Although the main focus of this study was on lncRNA genes, we also analyzed protein-coding genes in parallel. This approach was essential because lncRNA genes often act as regulators of other molecules, including proteins, and thus an analysis solely focusing on lncRNA genes would be incomplete. Furthermore, the original studies did not provide an accessible database for protein-coding gene expression profiles in the context of DOX effects and cardiotoxicity, a gap which our study effectively fills. Including protein-coding genes in our analysis further allowed us to confirm the findings of the original studies and increase the confidence in our results. However, it is worth noting that the data analyzed in this study originated from different laboratories and were based on different protocols, which may have introduced variability in the results. We tried to minimize this by using the same analysis pipeline for all four datasets, which were analyzed independently. One limitation is that the dysregulated lncRNA genes identified were limited to poly-A RNAs as all RNA-seq data were based on poly-A sequencing. This likely resulted in an underestimation of DOX-induced lncRNAs as more than half of them do not have poly-A tails [107]. Additionally, the validation experiments were carried out in vitro, which may not fully reflect the in vivo environment.

To facilitate further research in cardio-oncology, we constructed a comprehensive web database, DoxoDB, which enables researchers to explore expression profiles of both protein-coding and lncRNA genes in DOX-treated cells compared to control cells. Additionally, DoxoDB contains detailed information about the identified differentially expressed lncRNA genes, facilitating further experiments. Moving forward, to ensure that DoxoDB remains a reliable and up-to-date resource, the database will be regularly updated and monitored for users’ feedback. Furthermore, in our efforts to maintain DoxoDB’s relevance, we are open to incorporating additional datasets, which will enable researchers to explore a broader range of lncRNA candidates and enrich their investigations into the adverse effects of DOX. Given the flexibility and accessibility offered by DoxoDB, we anticipate that it will be a valuable resource for researchers looking into DOX adverse effects and for future functional and mechanistic studies investigating potential lncRNA candidates.

## 4. Materials and Methods

### 4.1. RNA-Seq Data Analysis and Visualization

As previously performed [43,108], all the RNA-seq data were downloaded from the Sequence Read Archive (SRA) and converted to FASTQ files using the SRA toolkit [109]. The FASTQ files were processed with fastp [110] (version 0.23.2), using default options, to perform quality control and reads filtering, reads pruning, and adapters trimming. The trimmed reads were aligned to the reference genome (GRCh38.109) using STAR [111] (version 2.7.9a). To calculate CPM and derive differentially expressed genes, the R package edgeR was used [112] (version 3.40.2). For all the analyses, the threshold of 2-fold and FDR < 0.05 were used to identify differentially expressed genes.

The full list of commands and programs used in this study can be found on the GitHub repository “Analysis_of_Doxo_Studies” (https://github.com/Reb08/Analysis_of_Doxo_Studies accessed on 4 June 2023).

The violin plots and volcano plots were generated using the R package ggplot2 [113] (version 3.4.1). The overlapping genes were identified using the R package VennDiagram [114], UpSetR [115] and the webtool https://bioinformatics.psb.ugent.be/webtools/Venn/ (accessed on 15 April 2023). The significance of the overlaps identified with the Venn Diagrams was calculated with the hypergeometric test using the R function phyper, while the Fisher’s exact test was carried out using the fisher.test function in R. Finally, GO and KEGG pathway analyses were carried out on the Database for Annotation, Visualization and Integrated Discovery (DAVID) (version v2023q1) using the GOTERM_BP_DIRECT, GOTERM_CC_DIRECT, GOTERM_MF_DIRECT, and KEGG_PATHWAY categories [116]. The significantly enriched terms were selected based on DAVID’s calculated FDR < 0.05.

The correlation analysis between the dysregulated lncRNA genes and other genes was based on Pearson’s correlation test and CPM values. The expression profile of each lncRNA gene was used to compute the correlation with the expression profiles of all protein-coding genes. To account for multiple testing, the FDR method was applied. Top-correlating protein-coding genes were chosen based on FDR < 0.001. Finally, lncRNA annotation was carried out using the information available from LncBook 2.0 [106]. The relevant LncBook 2.0 IDs were selected by matching the lncRNA coordinates from the GRCh38.109.gtf file to the coordinates from LncBook 2.0 (LncBookv2.0_GENCODEv34_GRCh38.gtf). Then, the obtained IDs were annotated based on LncBook 2.0 information on conservation (conservation_LncBook2.0.csv), miRNA-binding sites (lncRNAs_mirna_miRandaAndTargetScanAndRNAhybrid_LncBook2.0.csv), and GWAS traits (variation_LncBook2.0.csv). In addition, the R package GenomicRanges [117] was used to determine the nearest gene for each dysregulated lncRNA. The Pearson’s correlation test was used to determine the top correlated gene.

### 4.2. Cell Culture and Treatments

Human CFs (Innoprot (Derio (Bizkaia), Spain), #P10452) were cultured in the growth medium containing Minimum Essential Medium Eagle (MEM, Sigma-Aldrich (Søborg, Denmark), #M2279) supplemented with 15% fetal bovine serum (Sigma-Aldrich, #F4135), 1% L-Glutamine solution (Sigma-Aldrich, #G7513), and 1% Penicillin–Streptomycin (Sigma-Aldrich, #P4333). The cells were cultured at 37 °C with 5% CO2. Doxorubicin hydrochloride (DOX, Sigma-Aldrich, #D1515-10MG) was dissolved in water and used at the final concentrations of 1, 10, and 100 nM. The images were taken with BioTek Cytation 1 Cell Imaging Multimode Reader (Agilent Technologies (AH diagnostics) (Tilst, Denmark)).

### 4.3. Isolation of Total RNA and RT-PCR

As previously performed [43], to isolate and purify the total RNA, the TRIzol Reagent (Thermo Fisher Scientific, Roskilde, Denmark, #15596018) was used following the manufacturer’s protocol. To synthesize the first-strand complementary DNA (cDNA), SuperScript IV VILO Master Mix with the ezDNase Enzyme (Thermo Fisher Scientific, #11766500) was used to digest the genomic DNA and reverse transcribe one μg of total RNA for each sample, which was diluted with DNase/RNase-free water to the concentration of 1 ng/μL after the reverse transcription reaction. Using 1 ng of cDNA template per reaction, a quantitative reverse transcription polymerase chain reaction (qRT-PCR) reaction was performed with PowerUp SYBR Green Master Mix (Thermo Fisher Scientific, #A25777) via the QuantStudio 6 Flex Real-Time PCR System (Thermo Fisher Scientific) with the annealing temperature at 60 °C. Relative fold expression was calculated by 2^−DDCt^ using ribosomal protein lateral stalk subunit P0 (*RPLP0*) as an internal control. The primer pairs were designed using Primer3 (http://bioinfo.ut.ee/primer3-0.4.0/ accessed on 6 October 2022) [118] and in silico validated with the UCSC In Silico PCR tool (https://genome.ucsc.edu/cgi-bin/hgPcr accessed on 6 October 2022) before extensive testing by the conventional RT-PCR reaction followed by running the PCR product on an agarose gel to examine for a single band of the expected size for each primer pair. The primer sequences are provided in Appendix A.

### 4.4. Transfection of siRNAs

To silence *MAP3K4-AS1*, siRNAs were designed with RNAXS (http://rna.tbi.univie.ac.at/cgi-bin/RNAxs/RNAxs.cgi; accessed on 22 March 2023) and ordered from Sigma-Aldrich. The siRNA sequences are as follows: (#1) sense—AGGAACUGCUCCAUUCUAG[dT][dT], antisense—CUAGAAUGGAGCAGUUCCU[dT][dT]; (#2) sense—UGAGGUAAAACAAAACAAA[dT][dT], antisense—UUUGUUUUGUUUUACCUCA[dT][dT]; and (#3) sense—CUAGUUCUCUACUCGAUUU[dT][dT], antisense—AAAUCGAGUAGAGAACUAG[dT][dT]. Mission Negative control SIC-002, confidential sequence (Sigma-Aldrich), was used as control. Transient siRNA transfection (50 nM final concentration) was carried out using RNAiMax (Thermo Fisher Scientific, #13778150) according to the manufacturer’s protocol. The samples were collected three days after the transfection of siRNAs for the isolation of total RNA using RNeasy Mini Kit (Qiagen, Vedbæk, Denmark, #74106) and purified according to the manufacturer protocol.

### 4.5. Determination of Cell Viability and Apoptosis

CFs were plated in a 96-well plate with 10,000 cells per well. Next day, siRNAs were transfected. Two days after transfection of siRNAs, CFs were treated with 1 nM of DOX for one day. To assess cellular viability, cytotoxicity, and apoptosis, ApoTox-Glo Triplex Assay (Promega, Nacka, Sweden, #G6321) was used according to the manufacturer’s protocol. The fluorescent and luminescent signals were read via Varioskan LUX Plate Reader (Thermo Fisher Scientific).

### 4.6. Statistics of In Vitro Experiments

Data are presented as the mean + S.E.M. Microsoft Excel was used to calculate a *p*-value through the two-sample, two-tail, heteroscedastic Student’s *t*-test.

### 4.7. DoxoDB Web Application

The DoxoDB web application was created using the R package Shiny [119]. The app was designed with four main pages: (1) Explore, (2) lncRNAs, (3) Download, and (4) Documentation. These pages can be accessed through the navigation bar at the top. On the left-hand side of the Explore page, the main interactive Result table, rendered with the R package DT (https://github.com/rstudio/DT accessed on 6 April 2023), is displayed. This table shows the results of the RNA-seq data analysis and can be further explored on the five tabs on the right-hand side of the page through: (1)Volcano plot tab, displayed using the R package ggplot2 [113]; (2) Heatmap tab, rendered using the function pheatmap from the R package ComplexHeatmaps [120]; (3) GO analysis tab through the R package gprofiler2 [121]; (4) Pathway analysis performed with the enrichKEGG function from the R package clusterProfiler [122,123] to visualize the enriched KEGG pathways as a dotplot rendered with the R package enrichplot [124]; and (5) Comparisons Intersection tab through a Venn diagram, using the R package VennDiagram [114].

The annotations for the differentially expressed lncRNA genes (|log_2_FC| > 1 and FDR < 0.05) in each study and comparison can be found on the lncRNAs page. Here, the lncRNA Table, an interactive table rendered with the R package DT, can be accessed, and downloaded in tab-delimited value format (.tsv), along with the information displayed on the text boxes on the right had side of the page.

The datasets used in this study can be downloaded from the Download page in either comma-separated value (.csv) or .tsv format, where a preview of the data is displayed in a table rendered with the R package DT. Finally, instructions on how to use the web application and further information on the datasets can be found on the Documentation page.

All code used to generate DoxoDB is available in the GitHub repository: https://github.com/Reb08/DoxoDB/ (accessed on 4 June 2023). DoxoDB is freely available without password from https://rebeccadistefano.shinyapps.io/DoxoDB/ (accessed on 4 June 2023).

## Figures and Tables

**Figure 1 ncrna-09-00039-f001:**
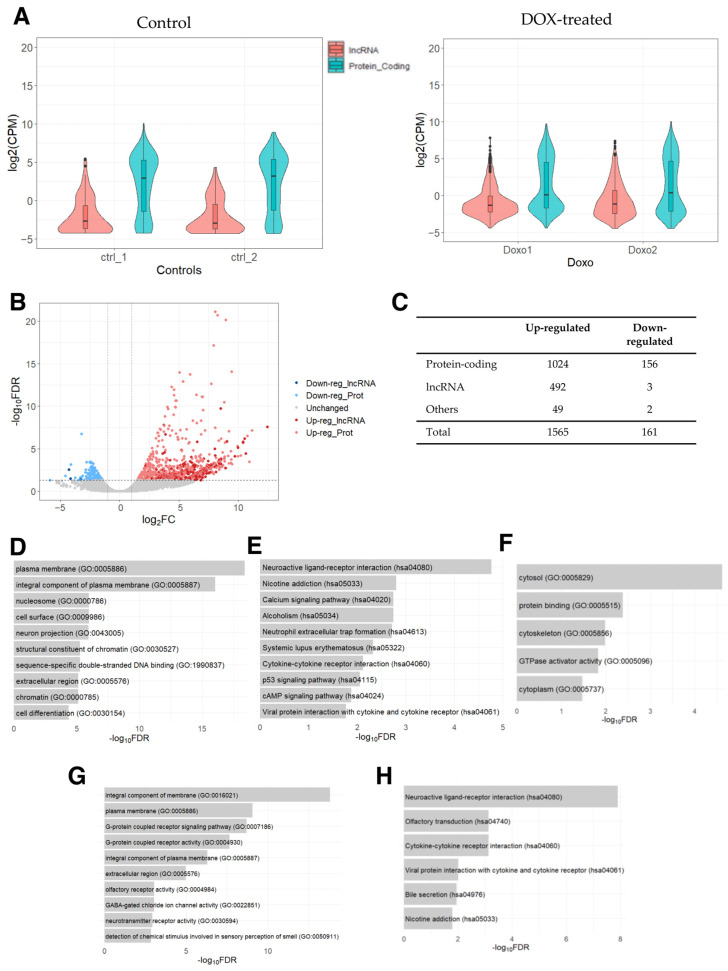
RNA-seq data analysis of DOX-treated PAAFs compared to the control cells. (**A**) Counts per million (CPM) values were used to draw a violin plot for each sample by distinguishing between protein-coding and lncRNA genes based on the annotation (biotype) provided by the Ensembl database (GRCh38.109). A small boxplot indicates the median and quartile range of expressions in CPM values. (**B**) Volcano plot of differentially expressed genes derived using 2-fold change and FDR < 0.05. (**C**) Number of differentially expressed lncRNA, protein-coding, and other genes (e.g., pseudogenes, rRNAs, miRNAs, and other small RNAs) based on the biotypes provided by the Ensembl database. (**D**) Top 10 enriched GO terms and (**E**) KEGG pathways for the list of up-regulated protein-coding genes. (**F**) Enriched GO terms for the list of down-regulated protein-coding genes. (**G**) Enriched GO terms (top 10) and (**H**) KEGG pathways for the top-correlated protein-coding genes in their expression to the set of up-regulated lncRNA genes.

**Figure 2 ncrna-09-00039-f002:**
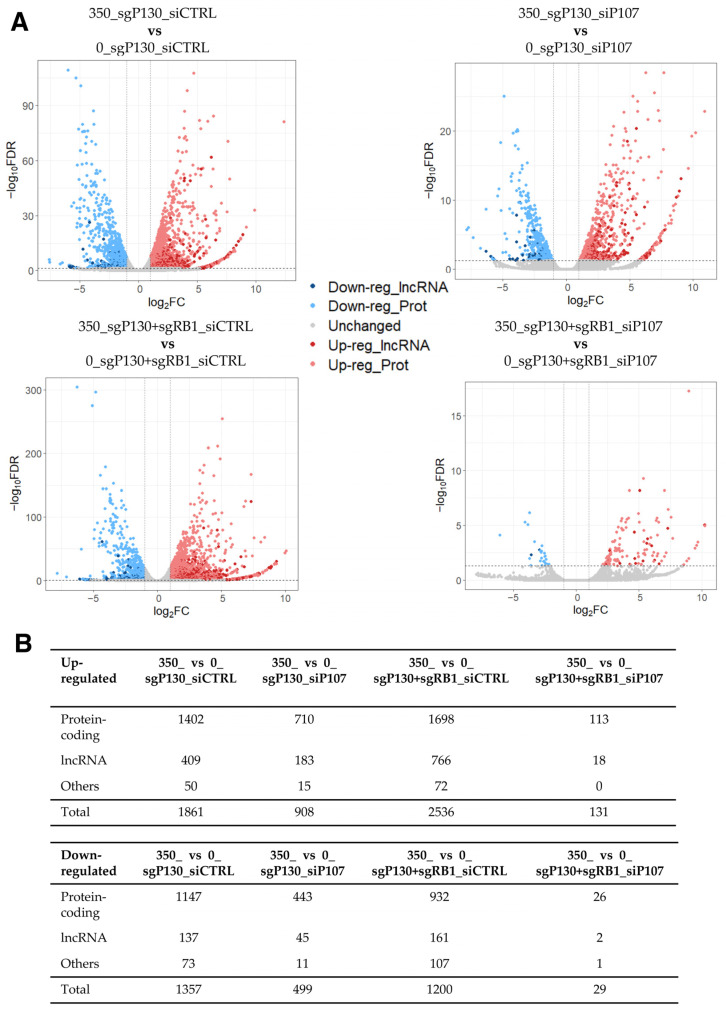
RNA-seq data analysis of DOX-treated sgP130_siCTRL, sgP130_siP107, sgP130+sgRB1_siCTRL, and sgP130+sgRB1_siP107 HFF cells compared to the control cells. (**A**) Volcano plot of differentially expressed genes among four comparisons. The threshold values of 2-fold and FDR < 0.05 were used to select differentially expressed genes. (**B**) Number of differentially expressed lncRNA, protein-coding, and other types of genes.

**Figure 3 ncrna-09-00039-f003:**
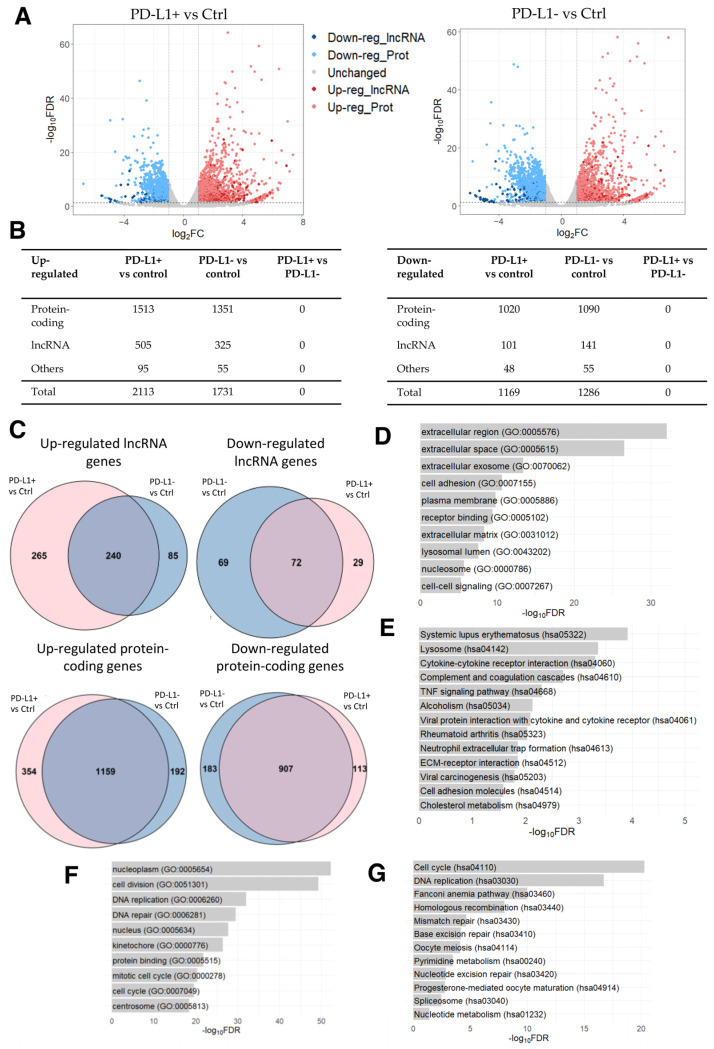
RNA-seq data analysis of DOX-induced senescent PD-L1+ and PD-L1- cells compared to the control cells. (**A**) Volcano plot of differentially expressed genes derived using the threshold values of 2-fold change and FDR < 0.05. (**B**) Number of differentially expressed lncRNA, protein-coding, and other types of genes. (**C**) Venn diagrams of shared up- and down-regulated lncRNA and protein-coding genes between DOX-induced senescent PD-L1+ and PD-L1- cells compared to the control cells (*p*-value < 0.0001). (**D**) Top 10 enriched GO terms and (**E**) KEGG pathways for the shared up-regulated protein-coding genes. (**F**) Top 10 enriched GO terms and (**G**) KEGG pathways for the shared down-regulated protein-coding genes.

**Figure 4 ncrna-09-00039-f004:**
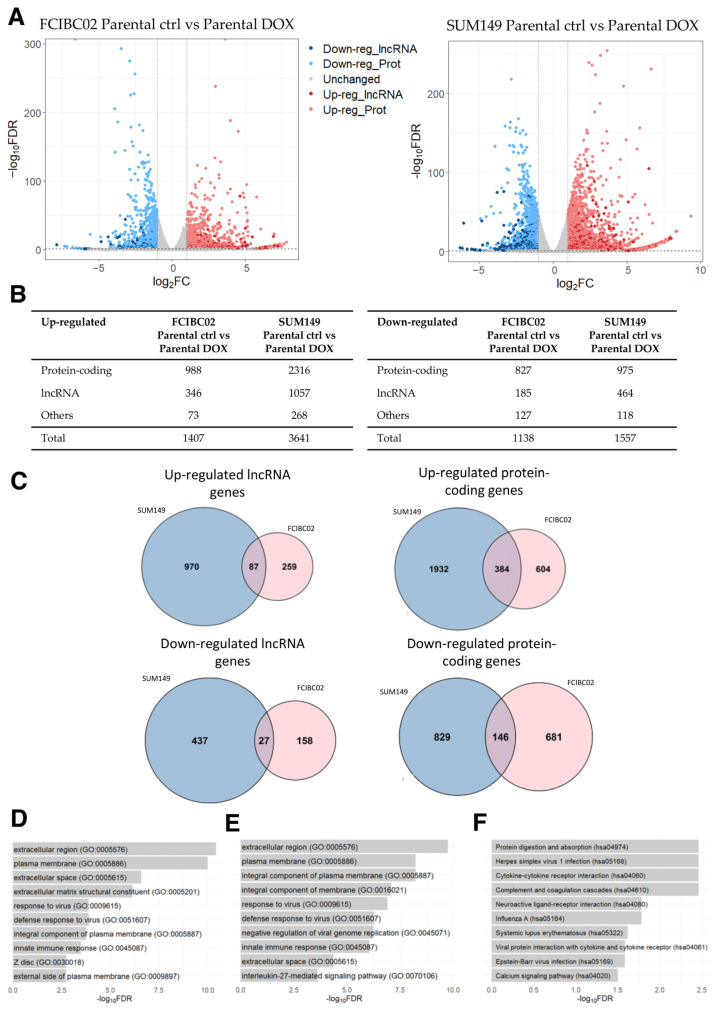
RNA-seq data analysis of DOX-treated IBC cells (FCIBC02 and SUM149) compared to control cells. (**A**) Volcano plot of differentially expressed genes. The threshold values of 2-fold change and FDR < 0.05 were applied. (**B**) Number of differentially expressed lncRNA, protein-coding, and other types of genes. (**C**) Venn diagrams of shared up- and down-regulated lncRNA and protein-coding genes between DOX-treated IBC FCIBC02 and SUM149 cells compared to the control cells (*p*-value < 0.0001). (**D**) Top 10 enriched GO terms for the shared up-regulated protein-coding genes. (**E**) Top 10 enriched GO terms and (**F**) KEGG pathways for the most significantly correlated protein-coding genes to the shared up-regulated lncRNA genes.

**Figure 5 ncrna-09-00039-f005:**
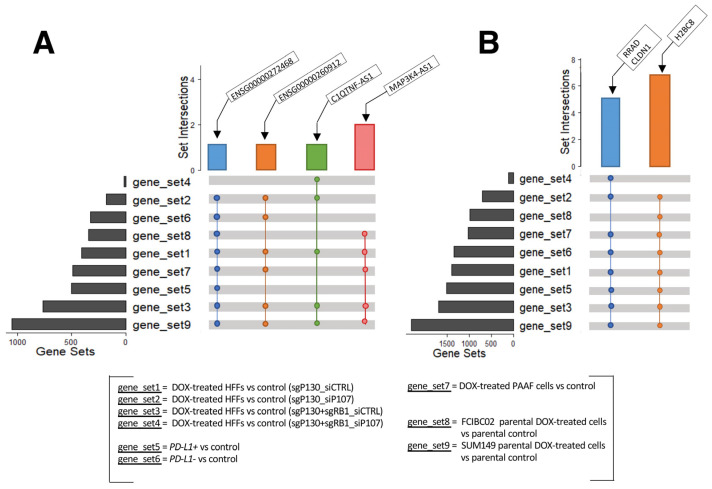
UpSet plots of shared genes. (**A**) Up-regulated lncRNA genes and (**B**) protein-coding genes shared among the different datasets analyzed (GSE154101, GSE135842, GSE198396, and GSE163361).

**Figure 6 ncrna-09-00039-f006:**
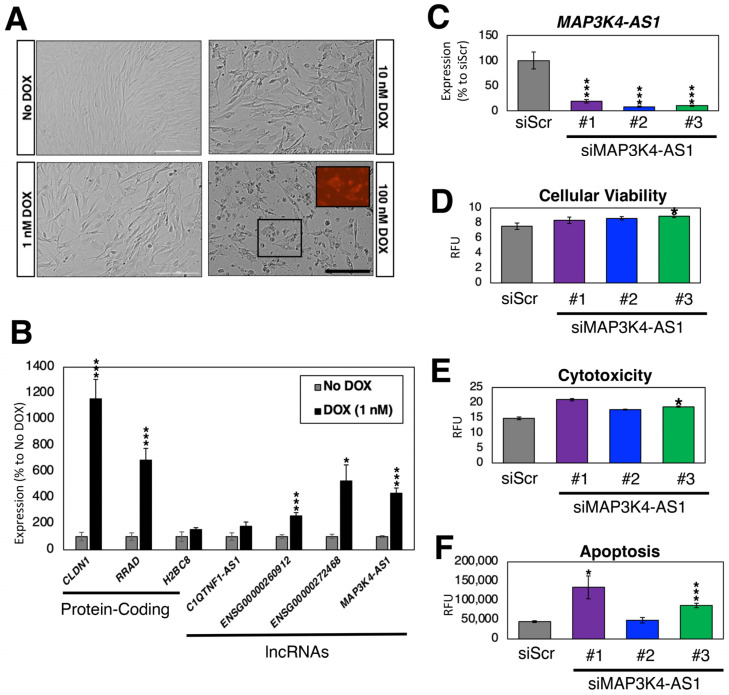
DOX treatment of CFs. (**A**) Bright field images of DOX-treated CFs after 24 h. The size of the scale bar is 200 μm. The inlet image in 100 nM DOX sample shows intrinsic fluorescence by DOX. (**B**) Expression profiles of DOX-regulated genes. *n* = 6 biological replicates. * (*p* < 0.05) and *** (*p* < 0.005). (**C**) Expression of *MAP3K4-AS1*. Compared to the control siRNA against scrambled sequence (siScr), all three siRNAs against *MAP3K4-AS1* showed statistically significant down-regulation of *MAP3K4-AS1*. *n* = 6 biological replicates. (**D**–**F**) Triplex assay to measure (**D**) cellular viability; (**E**) cytotoxicity; and (**F**) apoptosis measured by the activation of caspase-3/7. RFU stands for relative fluorescence units. *n* = 8 biological replicates.

**Figure 7 ncrna-09-00039-f007:**
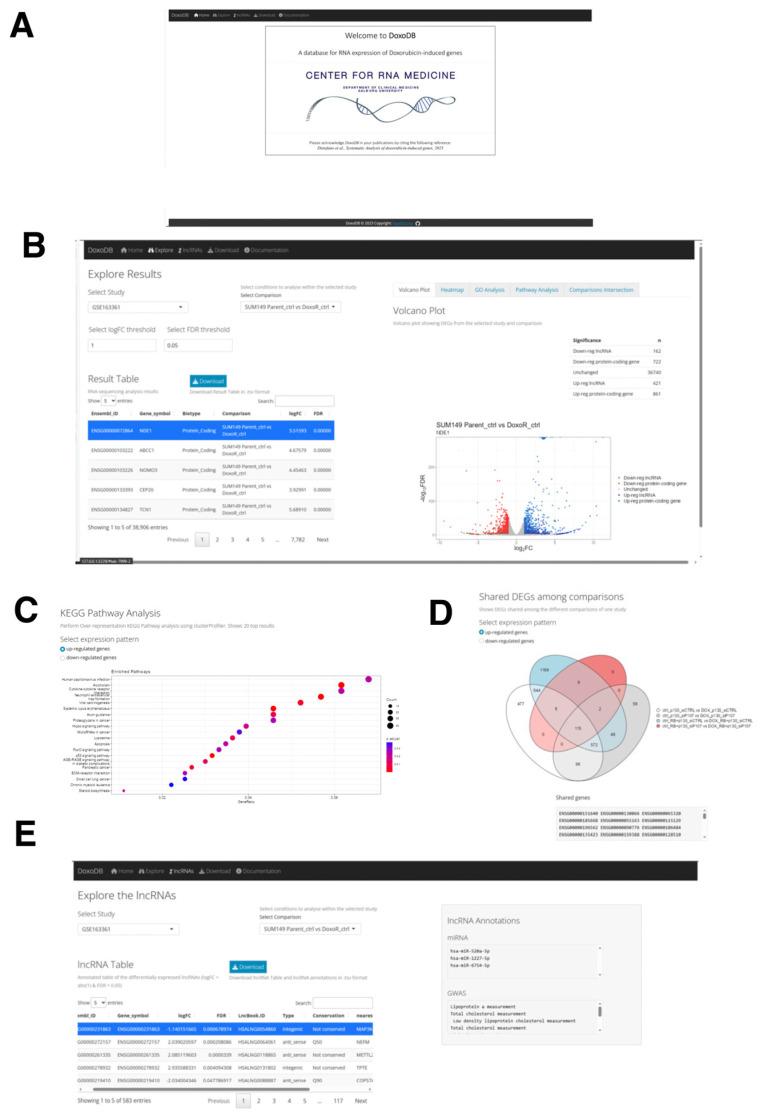
The DoxoDB web application. (**A**) Home page of DoxoDB. (**B**) “Explore Results” page. By setting the threshold values, the users can select the study and the condition to focus on. The results can then be dynamically visualized in the “Result Table” and in the adjacent Volcano plot. (**C**) KEGG pathway analysis of the DEGs. (**D**) Venn diagram to visualize the shared DEGs among the different conditions within the specified study. (**E**) “Explore the lncRNAs” page. The users can dynamically explore each differentially expressed lncRNA gene in each study and comparison from the lncRNA Table.

## Data Availability

The Appendix A can be found on the GitHub repository: https://github.com/Reb08/Analysis_of_Doxo_Studies/ (accessed on 4 June 2023). All codes used to generate DoxoDB are available on the GitHub repository: https://github.com/Reb08/DoxoDB/ (accessed on 4 June 2023).

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
