# Peer review of "DoxoDB: A Database for the Expression Analysis of Doxorubicin-Induced lncRNA Genes"

_ncrna, 2023, doi:10.3390/ncrna9040039_

Round 1

Reviewer 1 Report

Distefano and colleagues have generated a database for doxorubicin (DOX)- induced genes named as DoxoDB by including 4 different publically available RNA-seq datasets. they claim that this tool can be used to investigate lncRNAs that are induced due to DOX treatment in cardiac fibroblasts (CFs). The DoxoDB tool is relevant for the 'cardio-oncology' scientific community but there are some concerns on the methodology and logic applied to generate this dataset, which are as follows:

1. The samples (fibroblasts of lung, foreskin and inflammatory breast cancer cells) differ in cellular function and physiology. Therefore, each of the RNA-seq data sets arise from a different cell type that authors actually used for analysis and to generate DoxoDB. Also the DOX treatment conditions have not been uniform between these cell types. Moreover, the authors finally aimed to use it for identifying lncRNA induced by DOX in CFs. can the authors please comment on validity of using this strategy to find relevant DOX-induced genes for CFs? Wouldn't these identified genes be also deregulated in other cell types and not be specific for CFs?

2. primer pair sequence of RPLP0 is not specific since it amplifies regions in chromosome 2 and 12- as checked by UCSC BLAT tool. please clarify.

3. forward primer of ENSG00000272468 does not show any binding to the human genome sequence using UCSC BLAT. please correct.

4. Maybe a better idea to add a scheme for all the different transcripts (described between line 415-443).

5. How long were the CFs treated with DOX for testing the different doses?

6. Did the authors also investigate effect on proliferation (apart from apoptosis) as they identified genes related to proliferation in the analysis between DOX and control groups.

7. Why did the authors specifically choose MAP3K4-AS1 out of the 4 lncRNAs for the siRNA experiments?

8. Another limitation of this study is that only 4 lncRNAs were tested in DOX treated CFs and only 1 out of them was tested with siRNAs for functional readout. Thus, can the authors also validate some of the downregulated genes (protein coding as well as lncRNAs) to strengthen their analysis further?

9. The authors created the DoxoDB as a tool for researchers- but what is not clear is that if other new datasets can be included/added to this DB eventually or if the analysis is always limited to the 4 gene sets that the authors have used in this study? Based on line 549- can other gene sets be added in future where potentially non-polyA lncRNAs exist in the gene set?

it is fine.

Reviewer 2 Report

In this manuscript, the authors conducted an analysis of four published RNA-seq datasets derived from various cell types treated with either vehicle or doxorubicin. Their objective was to identify protein-coding genes and lncRNA genes that exhibited changes in expression levels in response to doxorubicin treatment. They developed a database called DoxoDB to disseminate the results, which includes information on differentially expressed genes based on fold change and FDR cutoffs, as well as enriched GO terms and KEGG pathways that provide insights into the potential affected biological processes.

Overall, the analyses conducted in this study are standard and straightforward, and the web database appears to be well-designed for presenting the information. It has the potential to serve as a valuable resource for researchers interested in understanding the regulation of gene expression by doxorubicin. However, there is one concern regarding this manuscript. The study was intended to investigate gene expression changes in cardiac fibroblasts and their potential impact on the cardiac system in patients undergoing doxorubicin treatment. However, the cell types analyzed in this study have limited relevance to the cardiac system, which creates a disconnect between the study's aims and the obtained results. It is suggested that either an RNA-seq dataset from cardiac fibroblasts be included or that the emphasis on the cardiac system be reduced.

Reviewer 3 Report

This work addresses an important aspect of cardio-oncology by investigating the adverse effects of the widely used chemotherapy drug doxorubicin. The authors performed their systematic analysis of published RNA sequencing (RNA-seq) data from doxorubicin-treated cells, which led to the identification of several doxorubicin-induced genes, including lncRNAs. This approach demonstrates the utilization of existing data to gain insights into the drug's effects. Furthermore, the authors conducted experiments on human cardiac fibroblasts treated with doxorubicin to validate their findings and performed a loss-of-function experiment on the lncRNA MAP3K4-AS1. These experimental procedures enhance the credibility of their results.

Additionally, the authors developed the web database DoxoDB to disseminate the analyzed data. This resource is valuable for the scientific community as it allows easy access to the expression profiles of doxorubicin-induced genes. However, further details regarding the features and functionality of the database could be provided to enhance its description.

Overall, the paper presents a comprehensive and well-executed study that contributes to the understanding of doxorubicin-induced gene expression changes, particularly in the context of lncRNAs. The findings are supported by both analysis of existing data and experimental validation. The development of the DoxoDB database further enhances the impact of the research.

Round 2

Reviewer 1 Report

Thank you to the authors for responding to all questions.  Here are 2 follow up minor comments:

there is a typo from the new text -please correct line 541- "devasting" should be "devastating".

for line 410-438 : it will be easier to read through this information in a tabular format with all the transcripts and their details instead of the current text format. The corresponding references can also be included in the table itself.

-
